# Modification of Bitumen with Recycled PET Plastics from Waste Materials

**DOI:** 10.3390/polym14214719

**Published:** 2022-11-04

**Authors:** Yerzhan Imanbayev, Akkenzhe Bussurmanova, Yerdos Ongarbayev, Akmaral Serikbayeva, Serik Sydykov, Maxat Tabylganov, Anar Akkenzheyeva, Nurlan Izteleu, Zhansaya Mussabekova, Dauren Amangeldin, Yerbol Tileuberdi

**Affiliations:** 1Engineering Faculty, Yessenov University, 32 Microdistrict, Aktau 130003, Kazakhstan; 2Faculty of Chemistry and Chemical Technology, Al-Farabi Kazakh National University, Almaty 050040, Kazakhstan; 3Institute of Natural Sciences and Geography, Abai Kazakh National Pedagogical University, Almaty 050040, Kazakhstan

**Keywords:** recycled polymer waste, petroleum bitumen, modification, physical and mechanical characteristics, microscopic analysis

## Abstract

Nowadays in the world, due to the constant desire for recycling, many countries are considering the use of recycled plastics on roads. Modification of bitumen for roads in Kazakhstan is considered one of the most suitable and popular approaches. This paper presents the results of research on the modification of bitumen by recycled plastics from waste materials. The paper describes the details of the use of plastic waste as bitumen modifiers, with a specific focus on recycled plastics and how they can potentially be used to enhance bitumen performance and the road durability. The main physical and mechanical characteristics of the modified bitumen were determined after routine tests, penetration and plasticity, softening temperature, brittleness temperature on Fraas and microscopic analysis. The morphology of the modified bitumen was studied using scanning electron microscopy. The results confirm that the modified bitumen complies with the requirements for polymer-bitumen binder of Kazakhstani standards and is suitable for the production of modified bitumen by its physical and chemical characteristics.

## 1. Introduction

Plastic materials have played a particularly important role in improving human living standards over the past 50 years. Plastics are the basis for the innovation of many products in various sectors such as construction, healthcare, electronics, automobile construction, packaging and others. Due to the rapid growth of the world population, the demand for plastic products has increased significantly. For example, in 2017, Victoria in Australia produced 586,300 tons of plastics. Only 22.2% (130,000 tons) were recycled and 1.2% (7200 tons) were combusted for energy recovery and the remaining 76.6% (449,100 tons) were sent to landfills [1,2,3,4]. The United States Environmental Protection Agency reported that 34.5 million tons of plastics were generated in the United States in 2015. According to statistics in Europe, about 38% of plastic waste goes to landfills, of which 26% are recovered, and 36% are recycled to obtain alternative energy sources. This shows that the largest amount of plastic waste is still sent to landfills, which is a consequence of the lack of free space in them [5,6,7,8].

Biodegradation of plastic can take about billions of years. They degrade very slowly because the molecular bonds of hydrogen, carbon and several other elements such as nitrogen, chlorine and others make the plastic very long life. The continuous accumulation of plastic products in landfills can lead to serious environmental problems. Improper disposal of plastic waste has a negative impact on the environment and human health due to many reasons: polymers have increased fire hazard properties, in other words, when burned, they are difficult to extinguish and emit toxic substances; toxic substances from burned polymeric materials can migrate into the atmosphere, lithosphere and hydrosphere [9,10,11]. In this regard, many countries have introduced a ban on the disposal of used polymers, and began to recycle them. Therefore, it is necessary to develop new and efficient methods for recycling plastic waste, which will address issues related to environmental protection and the production of additional types of energy sources. Therefore, such tasks as the development of technology for the modification of road bitumen with recycled plastics from waste materials obtained by recycling, the creation and improvement of specialized equipment, the production of modified bitumen with desired properties depending on the application, as well as other issues are relevant.

In the road industry, the most suitable classes of polymers for bitumen modification are thermoplastic elastomers and thermoplastics. Thermoplastic plastics account for about 80% of all world plastics produced. Therefore, the main part of the waste is plastic waste based on polyethylene terephthalate (PET) and they are used as additives in road production, since it consists mainly of polyester. Today there are not so many works devoted to the use of PET in the road industry; in particular, in works [9,10,11,12,13,14] it is shown how the use of PET directly in asphalt concrete mixture improves the resistance of the mixture to rutting. The authors of [13] have shown studies of the addition of PET in the composition of bitumen binder. According to the results of the study, it was found that the addition of PET in the range of 2–10% effectively affected the deformative characteristics of bitumen. This study recommends a new method of recycling PET waste for their use as part of a bitumen binder.

By the year scientists and road pavement technicians are represents new techniques and methods to improve practices of construction and maintenance with a purpose to minimize damages on the current environment. It was found that the most suitable classes of polymers for modifying bitumen are thermoplastic elastomers and thermoplastics in the road industry. Research studies [14,15,16] in relation to using plastics on roads have found noticeable improvements in the roads physical and mechanical characteristics namely, tensile strength, water resistance, durability, and overall life span. Therefore, the main part of the waste is plastic waste based on polyethylene terephthalate (PET) and they find applications as additives in road production, since they consist mainly of polyester. In [17] it is shown how the use of PET directly in an asphalt concrete mixture improves the resistance of the mixture to rutting. The authors of works [18,19] show studies of the addition of PET in the composition of a bituminous binder. According to the results of the study, it was found that the addition of PET in the range of 2–10% effectively affected the deformation characteristics of bitumen. This study recommends a new way to recycle PET waste for use as a bituminous binder.

In past decade, the operating conditions of road bitumen in the road coating have made it possible to formulate some requirements for the polymers used, which are most suitable for obtaining polymer-bitumen binders with tailor-made properties. That means the polymer molecules should have a tendency to association and should be well and quickly distributed in the bitumen dispersion medium without degradation. Polymers must form a structural network in bitumen. This structural network must retain elasticity at temperatures down to minus 60 °C and strength at temperatures not lower than 60 °C.

The state policy of Kazakhstan in the field of waste management is defined in the Concept for the transition of the Republic of Kazakhstan to a “green” economy and is aimed at introducing separate waste collection, developing the waste processing sector with obtaining products from recycled materials with the attraction of investments, including through public-private partnerships. In Kazakhstan, from 1 January 2019, a ban on the disposal of plastic, waste paper, cardboard and paper and glass waste came into force. In this regard, this study used PET waste from plastic bottles, which is an urgent need to solve the problem of utilization [20].

The goal of the research work is to research and develop a technology for modifying road bitumen with polymer waste for the utilization of hazard plastic waste and improving the physical and chemical rheological characteristics of Kazakhstan bitumen.

## 2. Materials and Methods

Recycled plastics from waste materials based on plastic bottles was used as base material of research study. Recycling company LLP “BVB-Alliance Atyrau” provide granulated plastic waste. Polymer waste is obtained by mechanical recycling. In the mechanical recycling method the polymer mass heated up to 200 °C, by extrusion it is squeezed out through holes in threads form, then they are immediately placed in water for cooling; after that, the threads are cut into granules. The essence of this method is the mechanical grinding of plastic waste for the purpose of further heat treatment and obtaining high-quality raw materials. Average diameter of plastic waste is 2 mm. Thermogravimetric analysis and physical-mechanical properties of recycled plastics were determined and shown in Table 1 and Figure 1.

The plastic sample turned out to be thermally stable (TG). Since, the effective weight loss of the plastic sample began upon reaching 415 °C, and at a temperature of 460 °C there was a sharp weight loss, which continued up to 506 °C, until the amount of sample in the crucible ran out. In the temperature range of 415–506 °C, about 97% of the mass of plastic was lost in 4.5 min (Figure 1).

In this study, bitumen of the BND 100/130 brand produced by “JV CASPI BITUM” LLP (Kazakhstan) was used for the preparation of modified bitumen. Conventional properties of the neat bitumen 100/130 are shown in Table 2.

The test results showed that the actual values of road bitumen, such as the softening point is 44 °C, ductility at 25 °C is more than 150 cm, dynamic viscosity at 135 °C –352 mm^2^/s, flash point −282 °C, brittleness temperature on Fraas −24 °C and complies with the requirements of Kazakhstani standards (KST) 1373-2013.

### Method of Preparation of Laboratory Samples of Polymer Bitumen Binders

Scheme of the installation shown in Figure 2. The reactor with sample (1) heated by an electric furnace (2) up to 175–180 °C, added polymer wastes, SBS and mixed. High accuracy thermometer (3) connected to the temperature controller. The samples stirring by a stirrer (4) in air environment. The stirring speed is 1000 rpm for 3 h. The stirrer consists of a metal propeller (5), a propeller speed controller (6). The process temperature controlled by automatic temperature controller (7). To avoid overheating of the polymer bitumen binders samples, all temperature changes are observed on the display (8).

Physical and mechanical characteristics determined by Kazakhstani standards. The softening point determined by the “Ring and ball” method according to KST 1227. Penetration determined by a penetrometer according to KST 1226. Ductility also indirectly characterizes the adhesion of bitumen and is associated with the nature of its components. The ductility determined with an automatic ductylometer according to KST 1374. The brittleness temperature on Fraas determined on an ATX-04 apparatus.

Ready-made highly elastic polymer SBS-01-10 (styrene-butadiene-styrene) was used as a modifier to prevent delamination of polymer bitumen binders. When this modifier is added to bitumen, the polymer-bitumen mixture becomes softer and more flexible at low temperature and more viscous at high temperature. The composition of SBS polymer is shown in Table 3.

The stability of dispersed systems including bitumen, depends on the degree of affinity of the maltenes and asphaltenes phase. In the case of instability, systems tend to phase separation. These processes are especially visible during storage, pumping, modification and heating. So, for example, in bitumens with a high content of asphaltenes and paraffin-naphthenic hydrocarbons, during the preparation of polymer bitumen binder, separation of the product due to precipitation of the polymer phase can be observed. Plasticizers are often used to stabilize the system. Fully synthetic motor oil (Shell Helix Ultra 5W-40) were used as plasticizer. The main characteristics presents in Table 4.

The images of modified bitumens were taken using JEOL JSM-6490 LA low-vacuum scanning electron microscope at Satpayev University. The polymer bitumen binders are fractured with a sharp tool and spread on steel plates to have a regular surface and placed for investigation under the microscope.

## 3. Results and Discussion

The softening point is useful in establishing the homogeneity of components and indicates the tendency of a material to flow at elevated temperatures. As can be seen from Figure 3, with an increase in the amount of added PET polymer waste, an increase in the softening temperature is observed, which characterizes the hardness of the material. The red line in Figure means the requirements of the technical standards for polymer bitumen binders (PBB) 90. That is, according to the technical standard, softening point must be at least 51 °C and, according to the graphs above the red line, meets the requirements of the technical standard. Without the addition of SBS polymer and in the presence of polymer waste alone, bitumen is cured in bitumen modification (curve 1), the softening temperature at a concentration of plastic waste of 3% has the highest value than that of samples with SBS. SBS plays as plasticizer in this mixture. These data confirm the hypothesis of gradual compaction of road bitumen in the presence of SBS and PET polymer waste. Among the prepared samples, binder with a content of 2 and 3 wt.% of polymer waste meets the requirements of the technical standards for PBB 90.

The red line in Figure 4 means the requirements of the technical standards for polymer bitumen binders and according to the technical standard, penetration must be at least 40. As expected, penetration decreases with an increase in the content of polymer waste. Polymers of PET adsorb bitumen oils and form a separate dispersed phase, which leads to a decrease in the oil/asphaltene ratio (O/A), resulting in an increase in viscosity and a decrease in penetration. Polymer waste with a concentration of 1% has the maximum effect on penetration, which can be explained by its structure, that is, due to the highly condensed cross-linked structure of the polymer, it is largely able to adsorb the maltene part of bitumen, and consequently increase the viscosity. It can also be noted that there is a tendency for a further decrease in penetration; perhaps an optimal O/A ratio is achieved. Analyzing Figure 3 and Figure 4, it can be concluded that the softening temperature and penetration are interrelated indicators; with a decrease in penetration, an increase in the softening temperature is observed.

Figure 5 shows the ductility changes monotonously and reaches its minimum value at a 3% content of PET polymer waste up to 10 cm. This is explained by the fact that in the process of dispersion of polymer waste, resins and polyaromatic components are found in the volume of insoluble swollen polymer particles, which affect the value of the ductility grades of the binder. Among the prepared samples, binder with a content of 1, 2 and 3 wt.% of polymer waste complies with the requirements of the technical specifications for PBB 90 in terms of indicators. The red line in Figure means the requirements of the technical standards for PBB 90. That is, according to the technical standard, ductility must be at least 30 cm and, according to the graphs above the red line, meets the requirements of the technical standard.

The change in the brittleness temperature is explained by the addition of PET polymer waste, which is a mixed component (Figure 6). According to the technical standard, brittleness temperature should be no higher than −25 °C and above the red line, meets the requirements of the technical standard. The brittleness temperature of binders in the case of compounding with the SBS type has a more complex dependence, and an increase in the SBS concentration leads to the adsorption of the oil-resin component of bitumen, while polymer waste reacts poorly with bitumen components. The oil-resin component of bitumen with a large number of aromatic compounds promotes intensive inturgescence, dispersion and dissolution of the polymer [21]. Due to the strong physical adsorption of bitumen oil-resinous components with polymer waste, they tend to retain hydrocarbons and resins on the periphery of the surface of micelles, which leads to plasticity. The using of polymer waste in the asphalt concrete mixture improves the resistance to rutting. It is assumed that the resulting binder as a whole has greater cohesive strength and high adhesive properties, which contribute to improving the stability of the modified asphalt concrete mixture to shear and dynamic deformations.

It is noteworthy that with a change in the concentration of the polymer additive over the entire range under study, the change in penetration, softening point and ductility has a distinct trend, while the brittleness temperature “hung” within the average value in minus 20 °C. This is explained by the saturation of the colloidal system with inhibition of polymer waste coagulation processes. The resulting polymer-bitumen binder is advisable to use in climatic regions where there are no low negative earth temperatures (below minus 30 °C). Upgrade the recipe by introducing a plasticizer that can lower the breaking point. Therefore, further experiments were carried out by introducing a motor oil as plasticizer for reducing the cost of the obtaining product.

As can be seen from the Table 5, the increase in the amount of plasticizer up to 10 wt.% leads to an increase in the depth of penetration of the needle, a decrease in the softening point to 53 °C and a slight increase in extensibility. At the same time, the additive is more than 6 wt.% of the plasticizer does not give a noticeable change in physical and mechanical properties. For example, penetration increases, but softening point and extensibility remain at about the same level. This is explained by the fact that in the presence of a plasticizer in the system, it leads to the complete distribution of polymer waste in bitumen and an increase in strength.

### Morphology Analysis of Polymer Modified Bitumen

It has been established that polymer waste in the composition of the binder acts as particles of the polymer component, which carry out dispersion-elastic reinforcement of asphalt concrete. At the same time, the PET particles do not completely decompose and particularly soluble in hot bitumen [22], but are associated with the bitumen components by strong, but rather mobile chemical bonds and show their qualities already in the composition of the new material. In their composition, bitumen acts as a liquid or pseudo-liquid thermoplastic matrix, and polymer waste particles create an elastic power frame in the volume of the binder.

The polymer bitumen binder is considered as a composite material in which the matrix (medium) is bitumen, and the dispersed phase is the polymer. Such binders are superior in properties to the properties of bitumen and polymers taken separately. Electron microscopic studies of polymer bitumen binder show that with small amounts of polymer waste (1–2% by weight) in modified bitumen, it is able to dissolve in the low molecular weight part of bitumen—oils. With large additives, the polymer is distributed in bitumen in the form of separate, unrelated particles [23]. The effect of their action in the composition is similar to the effect of the filler. With additions of 3% or more, particles aggregate and coalesce (Figure 7A). At low polymer concentrations, the compositions can be considered as dispersion-strengthened. Such an effect is observed when the content of the dispersed phase is 3% by weight. At high concentrations of SBS polymer in bitumen, the compositions can be considered as fibrous or layered, which have increased strength and elasticity (Figure 7B). With the addition of 3% PET and 1% SBS, the dispersion process is not so strong, in Figure 7C, particles of polymer waste are visible. This can further cause microcracks in the bitumen matrix. The process of mixing bitumen with polymers of any chemical nature at high temperature proceeds in two stages: emulsification of the softened polymer in liquid bitumen and subsequent partial swelling or complete dissolution. The depth of the polymer dispersion process in bitumen, other things being equal, is determined by the chemical nature and molecular weight of the polymer, the chemical composition of bitumen, and the ratio of components in the mixture. It is known [24] that the degree of dispersion of such systems, other things being equal, is determined by the ratio of the viscosity of the components, as well as mutual solubility. In the case of thermodynamic incompatibility (insoluble or partially soluble) components, the particle size limit in the mixture depends only on the ratio of viscosities and mixing conditions, and the mixture at elevated temperature is an emulsion (Figure 7C). Increasing the concentration of polymer waste leads to the destruction and crushing of the bitumen structure into smaller particles, there is an intensive mixing of the components (Figure 7E). Polymers are made up of large and branched molecules. They are intertwined and weakly subject to thermal motion. In intensive mixing, bitumen components quickly penetrate into the polymer mesh, adsorbing bitumen oils and forming a dispersed phase. In the case of a 3% content of polymer waste, the probability of their coalescence (merging) increases, leading to phase reversal in the system. Thus, styrene-butadiene-styrene forms a continuous phase in bitumen when introduced in an amount of at least 3% by weight (Figure 7F). For mutually soluble components, the degree of dispersion of the system further increases due to the interaction of the components at the phase boundary. The presence of aromatic blocks in the structure of the styrene-butadiene-styrene polymer determines its affinity for petroleum bitumen containing a significant amount of aromatic compounds. As a result, the structure of bitumen modified with SBS-type polymer is fundamentally different from the structure of bitumen compositions with polymer waste. At a mixing temperature of 175 °C due to the dissolution of the polymer in petrolene, a homogeneous composition is formed. It is assumed that this provides an opportunity to increase the deformation resistance of asphalt concrete in a wide range of operating temperatures, and significantly increase the durability of coatings.

## 4. Conclusions

An optimal formulation of a polymer-bitumen binder has been developed that has improved performance characteristics compared to petroleum bitumen. The most optimal recipe for the preparation of polymer-bitumen binders by adding plastic waste contains 3 wt.% PET, up to 3% SBS for required by the Kazakhstan standard PBB 40 and PBB 90.

The advantage of the used polymer additive is the stage of polymer preparation—the process of joint chemical destruction of recycled PET in the presence of SBS to inhibit the delamination of polymer components in bitumen, due to which the problem of manufacturing high-quality homogenization of the product is solved.

Further research is planned on how these modified bitumens will behave in the composition of asphalt concrete.

## Figures and Tables

**Figure 1 polymers-14-04719-f001:**
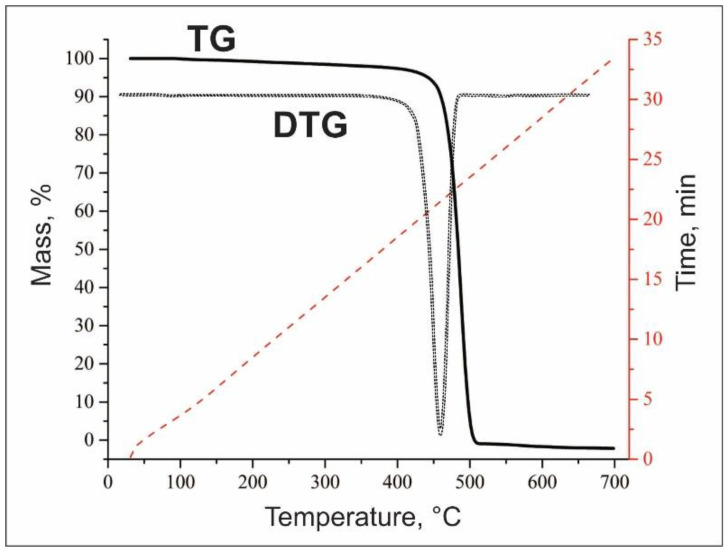
TGA of a plastic waste sample.

**Figure 2 polymers-14-04719-f002:**
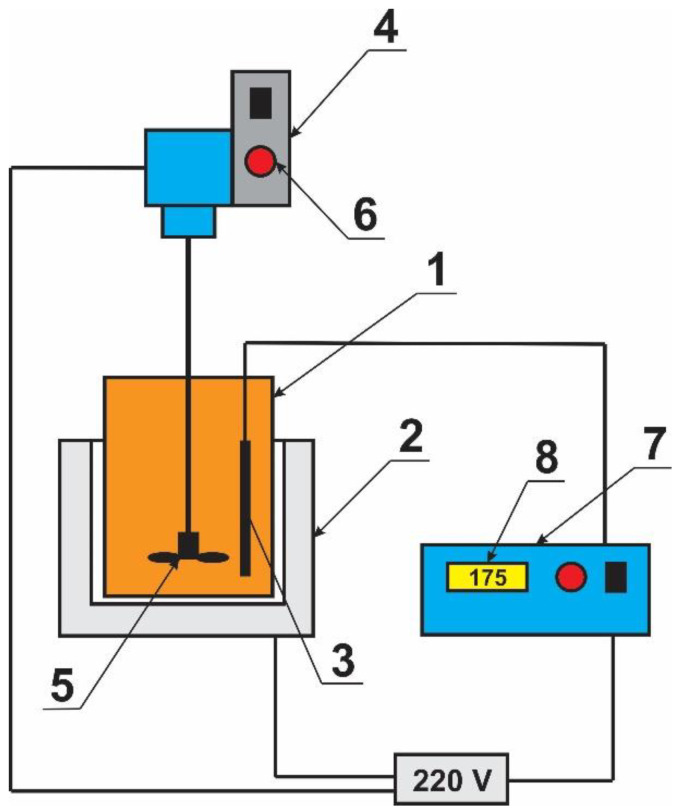
Scheme of the installation for the preparation of polymer bitumen binders.

**Figure 3 polymers-14-04719-f003:**
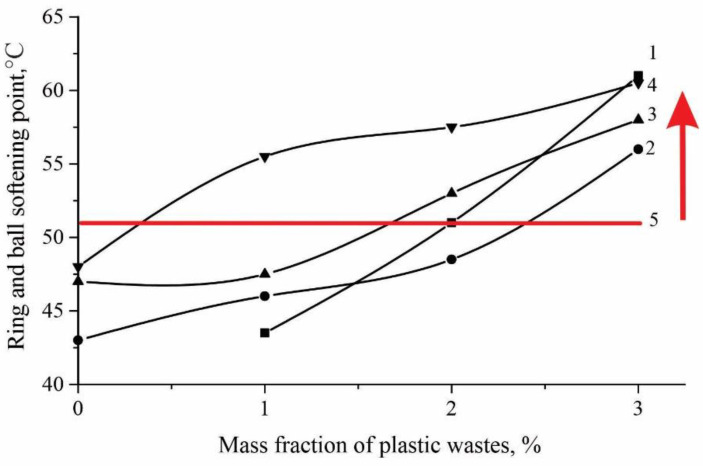
Change in softening temperature from polymer waste concentrations. 1—without SBS; 2—SBS 1%; 3—SBS 2%; 4—SBS 3%; 5—according to Standard.

**Figure 4 polymers-14-04719-f004:**
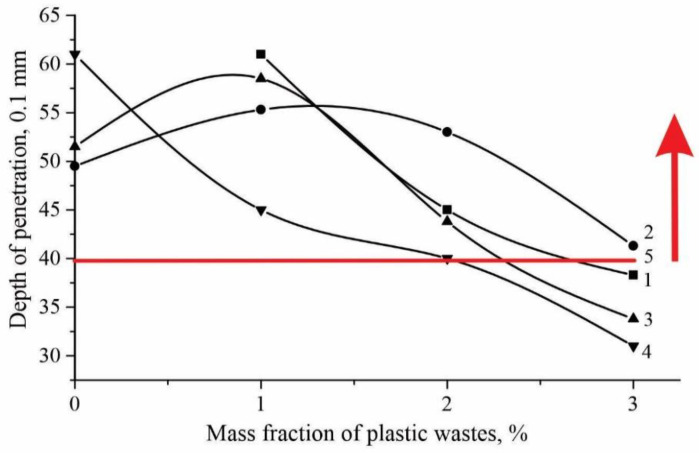
Change in penetrations from polymer waste concentrations. 1—without SBS; 2—SBS 1%; 3—SBS 2%; 4—SBS 3%; 5—according to Standard.

**Figure 5 polymers-14-04719-f005:**
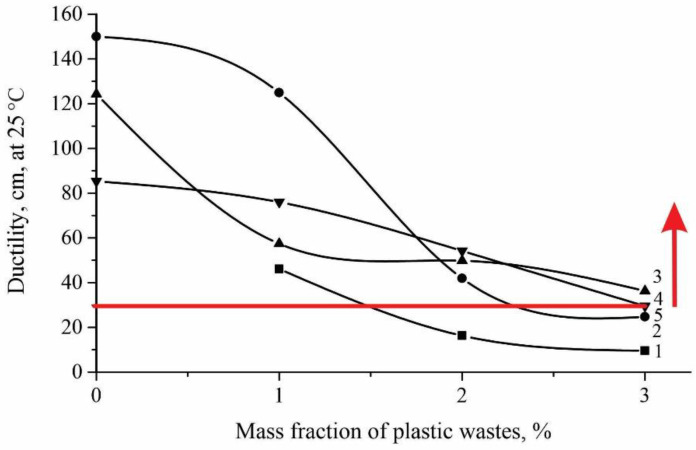
Change in ductility with polymer waste concentrations. 1—without SBS; 2—SBS 1%; 3—SBS 2%; 4—SBS 3%; 5—according to Standard.

**Figure 6 polymers-14-04719-f006:**
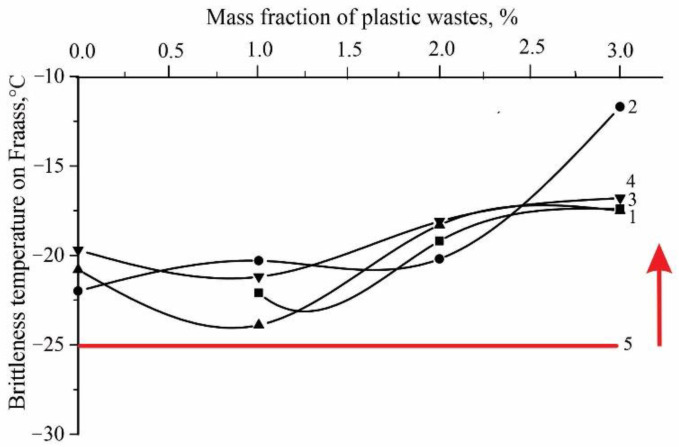
Change in brittleness temperature on Fraas from polymer waste concentrations. 1—without SBS; 2—SBS 1%; 3—SBS 2%; 4—SBS 3%; 5—according to Standard.

**Figure 7 polymers-14-04719-f007:**
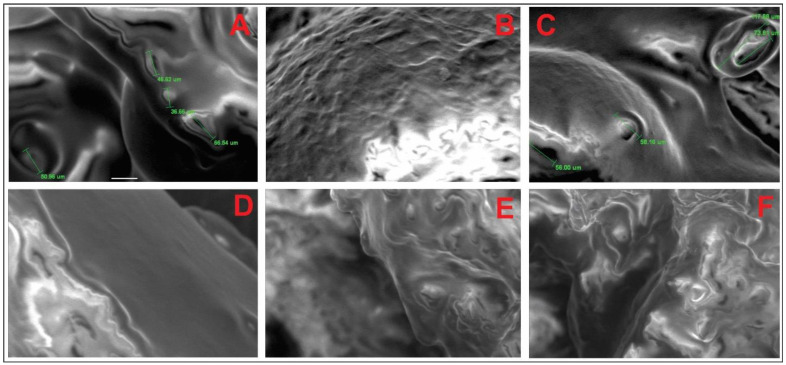
SEM images of modified bitumen. (**A**) Bitumen + 3% polymer waste. (**B**) Bitumen + 3% SBS. (**C**) Bitumen + 3% polymer waste + 1% SBS. (**D**) Bitumen + 1% polymer waste + 3% SBS. (**E**) Bitumen + 2% polymer waste + 3% SBS. (**F**) Bitumen + 3% polymer waste + 3% SBS.

**Table 1 polymers-14-04719-t001:** Physical and mechanical properties of recycled plastics.

Properties Indicator	Results
Tensile strength, MPa	8.9
Elongation at break, %	210
Frost resistance, °C	−45

**Table 2 polymers-14-04719-t002:** Characteristics of road bitumen 100/130.

Bitumen Properties	Normative Indicators of the Road Bitumen	Actual Value	Test Method
Penetration at 25 °C, not lower	101–130	113	Kazakhstani standards 1226
Softening point °C, not below	43	44	Kazakhstani standards 1227
Ductility at 25 °C, not less than, cm	90	>150	Kazakhstani standards 1374
Dynamic viscosity at 135 °C, mm^2^/s, not less	180	352	Kazakhstani standards 1210
Flash point °C, not below	230	282	Kazakhstani standards 1804
Brittleness temperature on Fraas °C, not higher	−22	−24	Kazakhstani standards 1229
Penetration index	From −0.1 until +1.0	−0.7	
Solubility %, not less	99.0	99.9	Kazakhstani standards 1228

**Table 3 polymers-14-04719-t003:** The composition of SBS polymer.

Polymer Properties	SBS-01-10
Structure	Linear
Bound styrene, %	30
Shore A hardness	80
Volatile matter, %	0.8
Ash content, %	0.3
Specific gravity	0.95
Tensile strength, MPa	21
Melt flow index, 200 °C/5 kgf	1

**Table 4 polymers-14-04719-t004:** The main characteristics of motor oil.

Properties	Shell Helix Ultra 5W-40
Density, kg/L	0.84
Pour point, °C	−39
Flash point, °C	215
Kinematic viscosity at 40 °C	74

**Table 5 polymers-14-04719-t005:** Physical-mechanical characteristics of polymer-bitumen binders with a content of 3% PET polymer waste and the addition of plasticizer.

The Name of Indicators	The Content of the Plasticizer, wt.%					Requirements of KST PBB 40
	0%	3%	5%	6%	10%	
Penetration at 25 °C	38	49	68	76	94	Not lower 40
Softening point, °C	61	58	56	53	54	Not lower 56
Ductility at 25 °C, cm	10	12	15	13	11	Not lower 15
Brittleness temperature on Fraas, °C	−17	−21	−21	−25	−31	Not higher −15

## Data Availability

Not applicable.

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
