# Peer review of "Modification of Bitumen with Recycled PET Plastics from Waste Materials"

_polymers, 2022, doi:10.3390/polym14214719_

Round 1

Reviewer 1 Report

This paper discusses the applicability of waste plastics to asphalt modifier, and it is considered to meet the needs of this times.

However, since waste plastics are limited to PET only, the scope of application become very narrow. In particular, the importance of diversion of PET bottles used as samples to asphalt modified materials is extremely low because mechanical recycling of bottle-to-bottles and chemical recycling to return them to monomers have been put into practical use by many companies. Therefore, it is necessary to expand the scope of application and consider the possibility of application from PET fibers from discarded clothing.

In addition, since the sample adjustment temperature is below the melting point of PET, it is very likely that PET itself has not melted. In such cases, PET is likely to function as a filler, in which case the properties will be highly dependent on the size of the initial addition period.

In addition, the following improvements are needed in the text.

In this study, there is no analysis of the characteristics of the PET itself, and there is a lack of an index of what kind of PET is good.

The composition of SBS is unknown.

OIL composition, chemical structure, etc. are unknown.

- In the graph, multiple lines intersect, making it very difficult to see. It should indicate what each symbol meets the sample.

In addition, a reference line is drawn on the graph, but it is unclear whether the upper is better or the lower is better, and it is difficult to judge.

Author Response

Response to comments from Reviewer to the manuscript

Modification of Bitumen with Recycled Plastics from Waste Materials”

Authors gratefully acknowledge the carefully critical review of the present manuscript. According to your comments submitted reconsidered manuscript and entered appropriate amendments and additions.

Here is a list of changes and corrections:

Reviewer 2 Report

The manuscript shows the results of the modification of bitumen with plastic waste. Although the manuscript shows a helpful approach to avoid further sending plastic waste to landfills, my primary concern is that the work lacks innovation (or at least, in the current manuscript, the innovation of the work is not clearly stated). Moreover, another major issue with the current text is that it does not show any characterization of plastic waste, which I consider essential in any scientific manuscript that describes the preparation of composite materials.

Other considerations:

Introduction section:
State of the art is not clearly described. The current text is confusing and does not follow a logical structure. Please rewrite.

Thermoplastics make up about 80% of all plastics produced. This phase appears two times in the introduction section.

What is the provenience of plastic waste? Was it municipal plastic waste, or did a recycling company provide it? What do authors mean by mechanical recycling? Please clarify this part.

Waste material, by definition, may not have constant composition. So, recycled plastic from waste materials can be formed by “pure” PET plastic waste or a mixture of different plastics (PE, PS, PP, PET, and so on). So, characterization of the plastic waste should be included in the manuscript (FTIR, thermal analysis). This request is made because the changes in the plastic waste composition may affect the properties of modified bitumen. Indeed, in Section 3 authors state that “The depth of the polymer dispersion process in bitumen, other things being equal, is determined by the chemical nature and molecular weight of the polymer, the chemical composition of bitumen, and the ratio of components in the mixture.” Then, this sentence highlights the importance of a physical-chemical characterization of plastic waste.

After mechanical grinding, what is the granulometric distribution of plastic waste? How did the granulometry used to prepare the plastic/bitumen mixtures was selected? Can it be expected that the granulometric distribution of plastic waste impacts the properties of the modified bitumen?

Did the authors consider or design any characterization protocol to verify the properties of the plastic waste before its mixture with bitumen?

Section 3 should be named “Results and Discussion”.

Author Response

(The authors gave the same response as above.)

Reviewer 3 Report

This publication discusses the addition of a polymer filler along with waste plastics to bitumen to act as rheological binders. The goal is in the application of these bituminous binders in concrete for road-laying applications. The authors evaluate microstructure (using SEM) and analyze performance using indicators such as brittleness temperatures, ductility, etc.

Formatting/non-technical suggestions

Overall, the paper can benefit from a closer look at the grammar and sentence constructions to make the reading experience smoother. Sentences like - " A slight decrease in the brittleness temperature is explained by the fact that the added polymer waste, acting as a mixed component, the waste is an agent that increases the brittleness temperature."  - are difficult to make sense of.  

The paper can also benefit from an improvement in presentation of the results and discussion.  A suggestion would be to present the SEM results first and then discuss the property enhancements in terms of the observed microstructure. 

The properties of the bitumen measured without any addition of plastic waste or SBS are presented in table 1. My suggestion would be to plot these values on the corresponding plots in figure 2-5. 

Scientific questions

The penetration results on page 6 show that the softening temperature increases, penetration reduces, and ductility also reduces with increasing plastic waster content. These properties although interrelated and controlled by the dynamics of the constituent materials show different trends sensitive to the specific composition of the mixture. Please discuss more in terms of what the relationship between these material properties is and how it is affected by the composition.

Figure 2 shows addition of SBS + plastic waste hardens the bitumen but sample 1 without adding the SBS is strengthened just by addition of plastic waste. the last data point of curve 1 (without any SBS) is higher than any other mixture. Please explain. 

Please explain - "In intensive mixing, bitu-men components quickly penetrate into the polymer mesh, pushing the chains apart and leading to a partial polymerization reaction." It is not obvious to me how this fits the observations.

Author Response

(The authors gave the same response as above.)

Round 2

Reviewer 1 Report

The revision is not sufficient because it does not fully answer the question.

Author Response

Response to comments from Reviewer to the manuscript

Modification of Bitumen with Recycled Plastics from Waste Materials”

Authors gratefully acknowledge the carefully critical review of the present manuscript. According to your comments submitted reconsidered manuscript and entered appropriate amendments and additions.

Here is a list of changes and corrections:

Comment 1.

What do the arrow marks represent in the figures? It is not clear to me what they are pointing towards. Please elaborate in figure captions.

Response.

The red line in Figure means the requirements of the technical standards for polymer bitumen binders (PBB) 90. For example, according to the technical standard, softening point must be at least 51 °C and, according to the graphs above the red line, meets the requirements of the technical standard.

Corresponding additions are added to the manuscript text.

Comment 2.

I would advise the authors to clearly sketch out their observations and supported conclusions in the results and discussions section. In the latest version of the manuscript, I am not able to follow the results and discussions section clearly. Some arguments are presented without any background evidence/reference while other observations are mentioned but never explained. Sentences like - "Proof plasticity appears." do not make any sense.

Response.

Corresponding additions are added to the manuscript text.

Reviewer 3 Report

I commend the authors for their results and sincerely request them to rewrite/edit the results and discussion section to present their ideas more coherently. Please see my comments below:

1. What do the arrow marks represent in the figures? It is not clear to me what they are pointing towards. Please elaborate in figure captions.

2. I would advise the authors to clearly sketch out their observations and supported conclusions in the results and discussions section. In the latest version of the manuscript, I am not able to follow the results and discussions section clearly. Some arguments are presented without any background evidence/reference while other observations are mentioned but never explained. Sentences like - "Proof plasticity appears." do not make any sense. 

Author Response

Response to comments from Reviewer to the manuscript

Modification of Bitumen with Recycled Plastics from Waste Materials”

Authors gratefully acknowledge the carefully critical review of the present manuscript. According to your comments submitted reconsidered manuscript and entered appropriate amendments and additions.

Here is a list of changes and corrections:

 Comment 1.

PET can not melt at the temperatures authors did.

Response.

According to the research work (Laboratory Properties of Waste PET Plastic-Modified Asphalt Mixes. Authors: Nuha Mashaan, Amin Chegenizadeh and Hamid Nikraz. Recycling 2021, 6, 49.) and our microscopic analysis was found PET polymers is soluble in hot bitumen. However, waste polymer is not reacted with bitumen components due to stable composition.

Comment 2.

And there is not merit to mix PET to the high quarity asphalt.

Response.

Produced not modified bitumens show a number of disadvantages in practice: high thermal sensitivity (softening at high temperatures and brittleness at low temperatures), poor mechanical characteristics and low elasticity, susceptibility to aging. Kazakhstan experiences an extreme continental climate, with long, hot summers and cold winters. Thermal sensitivity does not allow conventional bituminous mixtures to behave well at both high and low temperatures, so they need to be modification. The use of polymer modified bitumen (PMB) provides a reduction in the total cost of road construction by 30%, an increase in the service life of the road surface by 2-3 times, as well as a reduction in noise on the road by 3-4 times. Due to the use of PMB, the average annual cost of maintaining roads is reduced by 55-60 %. The savings of the road budget of the Republic of Kazakhstan from the use of PMB should amount to several billion tenge per year.

The goal of the research project funded by Government is a research and development of technology for modifying petroleum bitumen with waste polymers for the utilization of plastic waste and possibility improvement of the physical-chemical rheological characteristics of modified bitumen.

Round 3

Reviewer 3 Report

Thanks for making changes to the manuscript. I would like to point out one error - " In the paragraph after Figure 7, at one place, Figure 6C has been mentioned. This is a typo, it should be Figure 7C."

Author Response

Response to comments from Reviewer to the manuscript

Modification of Bitumen with Recycled Plastics from Waste Materials”

Authors gratefully acknowledge the carefully critical review of the present manuscript. According to your comments submitted reconsidered manuscript and entered appropriate amendments and additions.

Here is a list of changes and corrections:

Comment 1.

Thanks for making changes to the manuscript. I would like to point out one error - " In the paragraph after Figure 7, at one place, Figure 6C has been mentioned. This is a typo, it should be Figure 7C."

Response.

Corresponding additions are added to the manuscript text.
